# An investigation of bubble resonance and its implications for sound production by deep-water fishes

**Mark W. Sprague** [1]*, **Michael L. Fine**[2], **Timothy M. Cameron**[3]

1 Dept. of Physics, East Carolina University, Greenville, NC, United States of America, 2 Dept. of Biology, Virginia Commonwealth University, Richmond, VA, United States of America, 3 Dept. of Mechanical and Manufacturing Engineering, Miami University, Oxford, OH, United States of America

\* spraguem@ecu.edu

**Data Availability Statement:** All relevant data are within the paper and its Supporting information files.

**Funding:** The authors received no specific funding for this work.

## Abstract

Although the continental slope and abyss comprise the largest habitat on earth, the absence of documented fish sounds from deep waters is striking. Fishes with sexually dimorphic muscles attached to their swim bladders suggests that sounds are likely used in male courtship on the upper, mid and lower continental slope. To investigate the effects of environmental extremes on fish sound production, the acoustic behavior of a driven bubble is examined. This study is also relevant to target strength of sonar returns from fish and hearing in auditory specialist fishes. A bubble is a classic, if imperfect, model for swim bladder behavior since the swim-bladder wall is an anisotropic viscoelastic structure responsible for rapid damping. Acoustic properties of bubbles–including far-field resonant frequency, damping factor, and quality factor–are calculated in warm and cold surface conditions and in cold deep-water (depths 1000 m, 2000 m, and 3500 m) conditions using parameters for oxygen and nitrogen, the dominant gases in swim bladders. The far-field resonant frequency and damping factor of a bubble increase with depth, and the scattering cross-section and quality factor decrease with depth. These acoustic properties scale with undamped oscillation frequency of the bubble and do not vary significantly due to gas type or temperature. Bubbles in the deep-water environments are much less efficient radiators of sound than bubbles near the surface because the far-field radiated power for the same excitation decreases with depth. A bubble at depth 3500 m has a 25 dB loss in radiated sound power compared to the same-radius bubble at the surface. This reduction of radiation efficiency in deep water likely contributes to the absence of fish sound recordings in those environments.

## 1 Introduction

Vibration of an underwater resonant bubble has been of interest in many fields including oceanography, sonochemistry, biomedical ultrasonics, metamaterials, and the use of ultrasound for industries involving liquid ceramics, metals and foodstuffs [1]. Our interest in the subject has been the relationship of the acoustic vibration of fish swim bladders to that of an

**Competing interests:** The authors have declared that no competing interests exist.

**Abbreviations:** $A$, force amplitude; $c$, sound speed in the liquid; $D_p$, thermal diffusivity of the bubble gas; $F_0$, amplitude of driving for on spring system; $f$, frequency; $f'$, normalized frequency $f/f_{und} = \omega/\omega_{und}$; $f_M$, Minnaert frequency corresponding to angular frequency $\omega_M$; $f_R$, resonant frequency of driven system corresponding to angular frequency $\omega_R$; $f_{und}$, frequency for undamped oscillation corresponding to angular frequency $\omega_{und}$; $f_\sigma$, far-field pressure resonance frequency corresponding to angular frequency $\omega_\sigma$; $K$, stiffness parameter; $k$, spring force constant; $l_{th}$, thermal diffusion length; $P_{gas}$, equilibrium pressure in the bubble gas; $Q_f$, quality factor; $R$, bubble radius; $R_0$, equilibrium bubble radius; $t$, time; $X$, thermal diffusion ratio $R_0/l_{th}$; $x$, spring displacement; $\beta$, damping factor; $\beta'$, normalized damping factor $\beta/\omega_{und}$; $\beta_0$, non-acoustic damping factor; $\beta_{ac}$, acoustic damping factor; $\beta'_{ac}$, normalized acoustic damping factor $\beta_{ac}/\omega_{und}$; $\beta_{th}$, contribution to damping factor $\beta$ due to thermal effects; $\beta'_{th}$, normalized thermal damping factor $\beta_{th}/\omega_{und}$; $\beta_{vis}$, the contribution to damping factor $\beta$ due to viscous effects; $\beta'_{vis}$, normalized viscous damping factor $\beta_{vis}/\omega_{und}$; $\Gamma$, complex polytropic index; $\gamma$, ratio of specific heats for bubble gas; $\Delta\omega$, half-power width of resonance; $\varepsilon$, dimensionless frequency $\omega R_0/c$; $\eta_s$, shear viscosity coefficient of the liquid; $\rho_{liq}$, liquid density; $\sigma_s$, scattering cross-section; $\sigma'_s$, normalized scattering cross-section $\sigma_s/4\pi R_0^2$; $\tau$, liquid surface tension; $\omega$, angular frequency of the driving force; $\omega_0$, frequency-dependent parameter related to the resonant frequency; $\omega_M$, Minnaert angular frequency for bubbles; $\omega_{nat}$, natural angular frequency for bubble oscillations; $\omega_R$, resonant angular frequency of driven system; $\omega_{und}$, angular frequency of undamped oscillation; $\omega_\sigma$, far-field pressure resonance angular frequency.

underwater bubble. Since the 1960s the pulsating underwater resonant bubble has been the dominant paradigm for a swim bladder in a sound field and has influenced paradigms governing sound production, hearing and sound scattering [2–4]. The pulsating resonant bubble is a monopole that produces an omnidirectional sound field and has a resonant frequency that decreases with size and increases with depth [5]. Previous work on swim bladders as bubbles has typically employed Minnaert's equation for bubble resonance [6]. A number of studies including cod and other fishes placed in an underwater sound field indicated bladder resonance [7–9].

A large number of fish species [10, 11] are capable of sound production via muscles that drive swim bladder motion [12]. Most of these species are known from shallow water, estuaries and coral reefs [11, 13–15], but the fish contributions to the soundscape from deep water remain a void. Classic work by Marshall [16, 17] employed the presence of muscles attached to the swim bladder as a surrogate for sound production. Marshall determined sonic fishes on the continental slope were primarily in the families Marcouridae (rattails or grenadiers) and Ophidiidae (cusk-eels). Recent work on the Neobythitinae, a cusk-eel subfamily, indicates robust sonic systems from the upper slope to depths of 5 km [18, 19]. Despite recordings of whale sounds from numerous SOSUS stations, there are currently no confirmed recordings of a fish sound that originate in waters below the continental shelf. A potential fish sound has been localized to 600 m to 700 m from a multi-hydrophone array [20], and sounds have been recorded in aquaculture tanks from two cusk-eel species that live between 50 m, and 800 m [21]. Other work on deep-water sounds has been exploratory but largely negative or theoretical [22–25].

Recent work primarily on the oyster toadfish *Opsanus tau* has suggested a forced rather than a resonant response from the swim bladder. The fundamental frequency of the courtship boatwhistle call is determined by the contraction rate of superfast muscles [26, 27]. The call damps rapidly after contraction ceases, and fundamental frequency does not vary with fish size [27, 28]. The call has a directional rather than omnidirectional radiation pattern [29], and bladder deflation does not affect auditory thresholds [30]. There are fishes that produce swim bladder sounds that decrease in frequency with fish- and therefore swim-bladder size, but the frequency is lower than predicted by the traditional bubble resonance equation [5, 31, 32]. In weakfish *Cynoscion regalis* lower frequencies in larger fish have been ascribed to longer twitch times for larger muscles [33]. In other cases a tendon or bone drives swim bladder oscillation so that frequency is likely determined by the driving frequency rather than bladder properties [31, 32, 34, 35].

The swim bladder of the toadfish and fishes in general is an inefficient radiator for a number of major reasons. First, following Boyle's law, the pressure inside the bladder matches ambient pressure in most known teleosts [36], although in minnows [37] the luminal pressure is slightly above ambient. Equal pressure results in a slack membrane, which would have quite different properties than a bladder inflated above ambient or a model like a balloon. Secondly, the bladder of most sonic fishes does not move as a monopole during contraction of sonic muscles. In the toadfish for instance, muscles on the sides of the bladder push inward during contraction [26, 27]. The increased pressure inside the bladder pushes the bottom outward [28] resulting in an inefficient quadrupole movement in which gas is shuttled within the bladder rather than exerting pressure on the environment. Finally, recent evidence indicates that the toadfish swim-bladder wall is an anisotropic viscoelastic structure made of layers of collagen and elastin that contains a high water content that is responsible for rapid damping [38] and properties distinct from the underwater bubble contained within it.

In order to establish limits of the environmental effects on bubble oscillation, we examine the properties of a driven bubble of nitrogen and oxygen gas at various radii in three deep-

water environments of increasing depth and two shallow water environments for comparison. Since fish swim bladders contain a mixture of nitrogen and oxygen (and lesser amounts of carbon dioxide and argon) [39], our modeling work includes both dominant gases. We use depths of 1000 m, 2000 m, and 3500 m for the deep-water environments and surface parameters varying in temperature for the shallow-water environments. The parameters in the 3500 m deep environment should be close to the extremes of exposure for sonic fishes. We then use experimentally derived results from the toadfish to examine the potential interactions of the swim bladder wall with the gas inside. Our analysis is more sophisticated than the standard bubble equation dating back to the 1930s [6] and provides additional information. Our analysis predicts the far-field resonant frequency, damping factor, and quality factor that should be useful to a number of fields in fish bioacoustics. A summary of acoustical effects of depth on swim bladder vibration and how the effects are likely to impact fish acoustic communication on the continental slope is included in the discussion.

## 1.1 Bubble dynamics

Ainslie and Leighton [1] provide relationships for bubble oscillations based on the work of many others [40–42]. In this study, we use the notation from Ainslie and Leighton [1] and bubble dynamics using damping factors, scattering cross-sections, and resonant frequencies from Ainslie and Leighton [42]. Since we are using bubble oscillations to understand swim bladder oscillations in fishes, we assume a zero flow rate into and around the bubbles.

Minnaert [6] derived an expression for the natural frequency of a bubble that is large on the thermal and Laplace scales [1]. This "Minnaert" angular frequency is

$$\omega_{\mathrm{M}} = \frac{1}{R_0}\sqrt{\frac{3\gamma P_{\mathrm{liq}}}{\rho_{\mathrm{liq}}}},\tag{1}$$

where $R_0$ is the equilibrium bubble radius, $P_{\mathrm{liq}}$ the pressure in the liquid, $\rho_{\mathrm{liq}}$ the density of the liquid, and $\gamma$ the specific heat ratio of the bubble gas. The "Minnaert" frequency is $f_{\mathrm{M}} = \omega_{\mathrm{M}}/(2\pi)$.

The equation of motion for a driven gas bubble is

$$\ddot{R} + 2\beta\dot{R} + K(R - R_0) = -\frac{A}{\rho_{\mathrm{liq}}R_0}e^{i\omega t},\tag{2}$$

where $R$ is the bubble radius, $\beta$ the damping factor, $K$ the stiffness parameter, $A$ the force amplitude, $\omega$ the angular frequency of the driving force, and $t$ time. We can compare this equation to the well-known equation of motion for a driven mass on a spring [43],

$$\ddot{x} + 2\beta\dot{x} + \omega_{\mathrm{und}}^2 x = \frac{F_0}{m}e^{i\omega t}.\tag{3}$$

In the mass-spring system $x$ is the displacement of the spring from equilibrium, $\beta$ the damping factor, $\omega_{\mathrm{und}} = \sqrt{k/m}$ the angular frequency of undamped oscillation, $k$ the spring force constant, $m$ the mass, and $F_0$ the amplitude of the driving force. (We use the notation $\omega_{\mathrm{und}}$ here where most texts use $\omega_0$ to avoid confusion with another parameter $\omega_0$ related to bubble oscillations defined below.) Comparing Eqs (2) and (3), the stiffness parameter $K$ is analogous to $\omega_{\mathrm{und}}^2$, the frequency of undamped oscillation. One important difference between a driven bubble and a mass-spring system is that in the driven bubble, the parameters $\beta$ and $K$ are frequency-dependent, while in the mass-spring system $\beta$ and $\omega_{\mathrm{und}}$ are not. Nonetheless, we

define the frequency of undamped oscillation for bubbles with

$$\omega_{\text{und}} = 2\pi f_{\text{und}} = \sqrt{K}, \tag{4}$$

where $f_{\text{und}}$ is the frequency for undamped motion corresponding to angular frequency $\omega_{\text{und}}$.

The scattering cross-section of a bubble is defined [1] as the ratio of the scattered power to the incident power for a plane wave incident on the bubble. Although a fish swim bladder is not responding to an incident plane wave, the scattering cross-section can approximate the acoustic radiation of swim bladder vibrations into the water. The scattering cross-section of a bubble is [see Eq. (85) in Ainslie and Leighton [42]]

$$\sigma_s = \frac{4\pi R_0^2}{\left(\frac{\omega_0^2}{\omega^2} - 1 - 2\frac{\beta_0}{\omega}\varepsilon\right)^2 + \left(2\frac{\beta_0}{\omega} + \frac{\omega_0^2}{\omega^2}\varepsilon\right)^2}, \tag{5}$$

where $\omega_0$ a frequency-dependent parameter related to the resonant frequency, $\beta_0$ the non-acoustic damping factor, and $\varepsilon = \omega R_0/c$ the dimensionless frequency (in which $c$ is the sound speed in the liquid).

The term $\omega_0$ in Eq (5), which contributes to $\sqrt{K}$ in Eq (2), is

$$\omega_0^2 = 3\text{Re }\Gamma\frac{P_{\text{gas}}}{\rho_{\text{liq}}R_0^2} - 2\frac{\tau}{\rho_{\text{liq}}R_0^3}, \tag{6}$$

where $P_{\text{gas}}$ is the equilibrium pressure in the bubble gas and $\tau$ the liquid surface tension. The parameter $\Gamma$ is the complex polytropic index, given by

$$\Gamma(\omega) = \frac{\gamma}{1 - \left\{\frac{(1+i)X/2}{\tanh[(1+i)X/2]} - 1\right\}\frac{6i(\gamma-1)}{X^2}}, \tag{7}$$

where $\gamma$ is the ratio of specific heats for the bubble gas. The frequency dependence in $\Gamma(\omega)$ is due to the parameter $X$, the thermal diffusion ratio given by

$$X = \frac{R_0}{l_{\text{th}}}, \tag{8}$$

in which thermal diffusion length $l_{\text{th}}$ is

$$l_{\text{th}} = \sqrt{\frac{D_p(R_0)}{2\omega}} \tag{9}$$

with $D_p(R_0)$ the equilibrium thermal diffusivity of the bubble gas. In terms of these parameters the bubble stiffness [Eq. (87) in Ainslie and Leighton [1]] is

$$K = \omega_0^2 + \frac{\varepsilon^2}{1+\varepsilon^2}\omega^2. \tag{10}$$

The (total) damping factor $\beta$ is

$$\beta = \beta_0 + \beta_{\text{ac}}, \tag{11}$$

where $\beta_{\text{ac}}$ is the acoustic damping factor, given by

$$\beta_{\text{ac}} = \frac{\varepsilon^2}{1+\varepsilon^2}\frac{\omega}{2}. \tag{12}$$

The non-acoustic damping factor $\beta_0$ is

$$\beta_0 = \beta_{\text{th}} + \beta_{\text{vis}}, \tag{13}$$

where $\beta_{\text{th}}$ is the contribution due to thermal effects and $\beta_{\text{vis}}$ the contribution due to viscous effects. The contribution to damping due to thermal conduction is

$$\beta_{\text{th}} = \frac{3P_{\text{gas}}}{2\rho_{\text{liq}}R_0^2\omega}\,\text{Im}\Gamma. \tag{14}$$

The viscous damping term in Eq (13) is

$$\beta_{\text{vis}} = \frac{2\eta_s}{\rho_{\text{liq}}R_0^2}, \tag{15}$$

where $\eta_s$ is the shear viscosity coefficient of the liquid.

In this study we examine the dependence of these parameters on water and gas properties in five environments: a warm surface environment; a cold surface environment; and cold deep environments at depths of 1000 m, 2000 m, and 3500 m. We also examine how these properties relate to fish swim bladders. Since the parameters in the bubble equations have a complicated frequency dependence, we use numerical techniques to solve for radiated sound and the resonant frequencies of the driven bubble system. Where appropriate, we use the same techniques used for the mass-spring system.

## 1.2 Resonance frequencies

Ainsley and Leighton [1, 42] describe several different resonance frequencies for bubbles, including the natural frequency for unforced motion, the displacement resonance, the velocity resonance, the bubble pressure resonance, and the far-field pressure resonance. The natural angular frequency for bubble oscillations is

$$\omega_{\text{nat}} = \sqrt{K - \beta^2}. \tag{16}$$

Since both $K$ and $\beta$ are functions of frequency [see Eqs (10)–(14)], the natural angular frequency is the solution to the equation

$$\omega_{\text{nat}}^2 = K(\omega_{\text{nat}}) - [\beta(\omega_{\text{nat}})]^2. \tag{17}$$

The resonant frequency most important for radiation of sound by fish swim bladders is the far-field pressure resonance with angular frequency $\omega_\sigma = 2\pi f_\sigma$ and frequency $f_\sigma$. This resonance is defined [1] as the frequency for which the scattering cross-section $\sigma_s$ is maximum. At this frequency the swim bladder radiates sound power into the far field most efficiently. We determine $\omega_\sigma$ values by substituting seawater and gas parameters into Eq (5) and numerically solving for the angular frequency at which the function is maximum.

## 1.3 Quality factor

The quality factor $Q_f$ of a resonance is a parameter that indicates the sharpness of its resonance curve (not to be confused with the $Q$ factors in Ainslie and Leighton [1], which represent different parameters). A high $Q_f$ value indicates a sharp peak of the resonance curve of the system response vs. driving frequency, and a low $Q_f$ value indicates a broad peak in the resonance

curve. Quality factor $Q_f$ is defined [43] as

$$Q_f = \frac{\omega_R}{2\beta},$$ (18)

where $\omega_R = 2\pi f_R$ is the resonant angular frequency, $f_R$ the resonant frequency, and $\beta$ the damping factor (at the resonant frequency). In this study we use $\omega_\sigma$ as the resonant frequency for the quality factor and $\beta$ as defined in Eq (11) for the damping factor. The quality factor of a lightly damped oscillator ($\beta \ll \omega_0$) has the familiar form [43]

$$Q_f = \frac{\omega_R}{\Delta\omega},$$ (19)

where $\Delta\omega$ is the width in angular frequency between the response function points at $1/\sqrt{2}$ times the maximum amplitude, corresponding to the half-power width of the resonance. This approximation reinforces the idea of the quality factor $Q_f$ as a measure of the width of the resonance peak.

## 2 Computational methods

In this study we use computational methods for analytical calculations with complicated terms as well as for finding numerical solutions of equations and for determining extrema of parameters. All calculations were done using the the Julia Programming Language [44]. Jupyter notebooks with the calculations are provided in Supporting Information (*cf.* S1–S4 Files). We use the Julia package SymPy, which calls the Python SymPy package [45] to manipulate large symbolic expressions, to make complicated substitutions, and to calculate derivatives analytically. If we can obtain a symbolic expression for a parameter, we convert the expression into a Julia-native function using the `lambdify` function from SymPy to speed calculations of large numbers of values and to use the expression in numerical methods. In most cases we are able to use double-precision floating-point (64 bit) calculations to determine values of quantities. When we determine that there is a significant loss of precision in the double-precision calculations, we use Julia `BigFloat` numbers with a default precision of 256 bits. We use the Julia package Roots [46] to find numerical solutions to equations with the secant method and the bisection method as implemented by the `find_zero` function. We use the Julia package Optim [47, 48] to numerically determine extrema of parameters with Brent's method [49] as implemented in the `optimize` function. A general procedure for calculations follows.

1. Use SymPy to make substitutions and to manipulate expressions analytically.

2. Convert the resulting SymPy expression to a Julia-native numerical function with the `lambdify` command.

3. If the expression is solved for the desired quantity, substitute parameter values to generate graphs and other results.

4. If the expression has no analytical solution, use the `find_zero` function in the Roots package to obtain numerical solutions for different parameter values.

5. If the desired parameter is a maximum or minimum value of an expression with no analytical solution, use the `optimize` function from the Optim package to determine the value numerically.

For some quantities, we calculate an array of values at evenly spaced parameter values and use interpolation to estimate quantity values for parameter values between those used for

calculations. We use the Julia Interpolations package [50] to perform a third-order cubic-spline interpolation as implemented in the `interpolate` function.

## 3 Water and gas properties

Several water and gas properties are necessary for calculation of bubble oscillations including water temperature, density, dynamic viscosity, surface tension, and sound speed, as well as the ratio of specific heats and thermal diffusivity of the bubble gas. We calculate water and gas properties for each of the environments and compare our calculated values with tabulated values over a large range of conditions to determine the numerical precision required in our calculations for accurate results (*cf.* S1 and S2 Files). The deep-water environments have a temperature of 1.5˚C, a typical water temperature at a depth of 3500 m [51]. The warm shallow-water environment has a temperature of 20.0˚C (typical conditions for shallow-water soniferous fishes), and the cold shallow-water environment has a water temperature of 1.5˚C. All environments have a salinity of 35.0 g/kg. The pressure in the deep-water environments is calculated with the pressure vs. depth relationship in Saunders and Fofonoff [52], and the pressure in the shallow-water environments is standard atmospheric pressure. We calculate the water density, dynamic viscosity, and surface tension for the environments with the relationships given in Sharqawy *et al.* [53] and the water sound speed with the UN equation [54]. We calculate the isobaric and isochoric heat capacities using the relationships given in Span *et al.* [55] for nitrogen and Schmidt and Wagner [56] for oxygen. We validated these calculations by comparing our calculated parameters to those in Span *et al.* [55] for nitrogen and Weber [57] for oxygen. From the heat capacities, we obtain the ratios of specific heats for the two gases. We calculate the thermal diffusivities of nitrogen and oxygen using the relationships given in Lemmon and Jacobsen [58] for thermal conductivity and then the relationship between thermal diffusivity, thermal conductivity, density, and isobaric specific heat as defined in Ainslie and Leighton [1]. Table 1 gives the seawater, nitrogen, and oxygen parameters in each of the environments. In order to show how bubble properties vary with depth, we also calculated each of these properties at 1 m depth intervals from the surface to a depth of 3500 m and used interpolation to estimate the values at intermediate depths (*cf.* section 4 in S3 and S4 Files).

**Table 1. Seawater, nitrogen, and oxygen parameters used to calculate bubble response curves and resonances.**

| Parameter | Surface | | Deep | | |
|---|---|---|---|---|---|
| | **Warm** | **Cold** | **1 km** | **2 km** | **3.5 km** |
| Depth (m) | 0 | 0 | 1000 | 2000 | 3500 |
| Temperature (˚C) | 20.0 | 1.50 | 1.50 | 1.50 | 1.50 |
| Ambient Pressure (MPa) | 0.101325 | 0.101325 | 10.19 | 20.33 | 35.62 |
| Water Density (kg/m$^3$) | 1028 | 1027 | 1032 | 1036 | 1043 |
| Water Dynamic Viscosity (mPa s) | 1.077 | 1.812 | 1.812 | 1.812 | 1.812 |
| Water Surface Tension (mN/m) | 73.52 | 76.01 | 76.01 | 76.01 | 76.01 |
| Water Sound Speed (m/s) | 1522 | 1456 | 1473 | 1490 | 1516 |
| $N_2$ Ratio of Specific Heats $\gamma$ | 1.401 | 1.402 | 1.612 | 1.738 | 1.760 |
| $O_2$ Ratio of Specific Heats $\gamma$ | 1.397 | 1.400 | 1.668 | 1.897 | 1.950 |
| $N_2$ Thermal Diffusivity ($10^{-9}$ m$^2$/s) | 21000 | 18620 | 191.4 | 114.3 | 95.38 |
| $O_2$ Thermal Diffusivity ($10^{-9}$ m$^2$/s) | 21210 | 18780 | 172.0 | 92.64 | 74.87 |

See S1 and S2 Files for the calculations of these values.

## 4 Bubble resonance

Considering the different resonance frequencies for bubbles, and that the gas in the bubbles affects the frequencies, we examine the differences in $\omega_{\text{und}}$ and $\omega_\sigma$ for both nitrogen and oxygen bubbles in terms of bubble size for the five environments.

### 4.1 Undamped frequency

We calculate values of the undamped bubble frequency $f_{\text{und}}$ using Eqs (4), (6) and (10) by numerically solving for the frequency at which $\sqrt{K} = \omega$ with specific values of gas and water properties and bubble radii (*cf.* section 5.4 in S3 and S4 Files). Fig 1(A) is a graph of $f_{\text{und}}$ vs. bubble radius for nitrogen and oxygen bubbles in each of the environments. Fig 2(A) is a graph of $f_{\text{und}}$ vs. depth for nitrogen and oxygen bubbles of radii 0.01 m, 0.05 m, and 0.10 m for a 1.5˚C constant temperature, 35.0 g/kg constant salinity depth profile. The $f_{\text{und}}$ values for nitrogen and oxygen bubbles are very similar, but smaller-radius, deeper-water bubbles have a greater frequency difference between oxygen and nitrogen bubbles than larger-radius or shallower bubbles.

### 4.2 Far field pressure resonance

We calculated values of the far-field resonance frequency $f_\sigma$ by maximizing the scattering cross-section $\sigma_s$ in Eq (5) numerically for specific values of gas and water properties and bubble radii (*cf.* sections 5.4 and 5.5 in S3 and S4 Files). Fig 1(B) is a graph of $f_\sigma$ vs. bubble radius for nitrogen and oxygen bubbles in each environment. There is very little difference between the far-field resonance frequencies for nitrogen and oxygen bubbles in each of the five environments. Furthermore, the far-field resonance frequencies for bubbles in the two shallow-water environments are virtually the same. At all bubble radii the values of $f_\sigma$ increase with depth. Also, the $f_\sigma$ values have a greater variation with bubble radius as the water depth increases.

Since there is little variation between the resonant frequencies due to temperature alone [see the surface curves in Fig 1(B)], we calculate the far-field bubble resonant frequency vs. depth for a 1.5˚C constant temperature, 35.0 g/kg constant salinity depth profile [Fig 2(B)] to show the effect of water depth on the resonant frequency (*cf.* section 6 in S3 and S4 Files). The

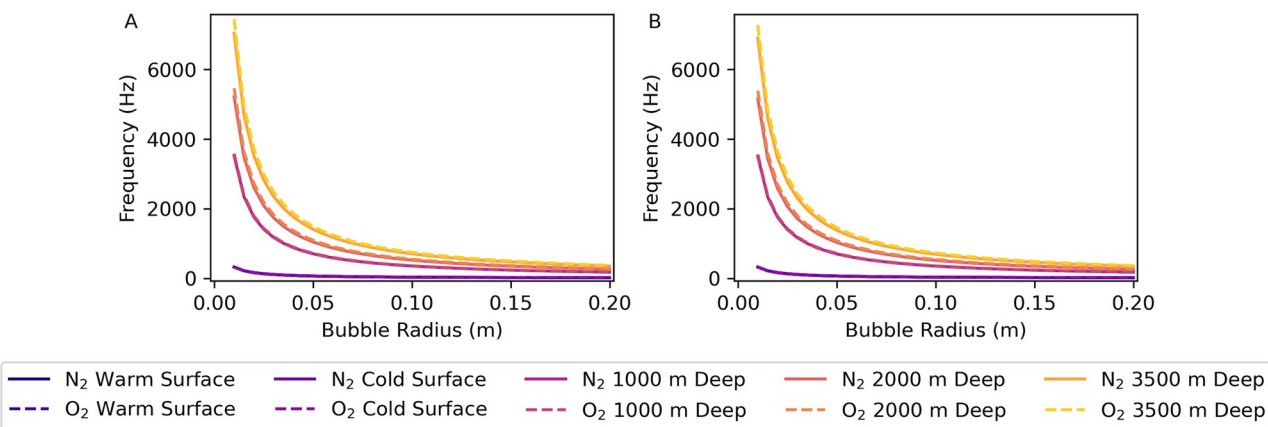

**Fig 1. Bubble frequency vs. bubble radius for nitrogen and oxygen bubbles.** (A) Undamped bubble oscillation frequency $f_{\text{und}} = \omega_{\text{und}}/(2\pi) = \sqrt{K}/(2\pi)$ vs. bubble radius. (B) Far field resonant frequency $f_\sigma$ vs. bubble radius. Parameter values for each environment are given in Table 1. See sections 5.4–5.7 in S3 and S4 Files for the calculations used for this plot. The curves for nitrogen and oxygen bubbles in the warm and cold shallow-water environments all overlap in both subplots A and B.

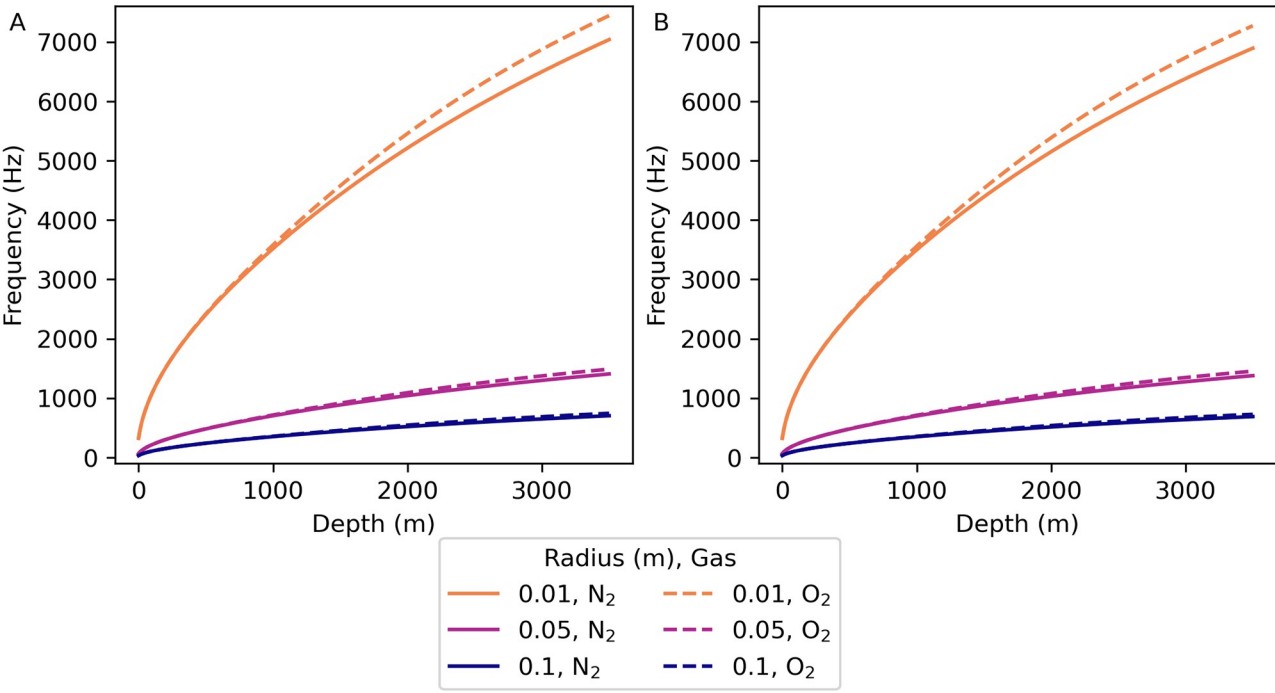

**Fig 2. Bubble oscillation frequencies vs. depth.** Frequencies calculated for nitrogen and oxygen bubbles of radii 0.01 m, 0.05 m, and 0.10 m for a 1.5˚C constant temperature, 35.0 g/kg constant salinity depth profile. Water and gas parameter values were calculated at 1 m depth intervals and interpolated between those depths. (A) Undamped bubble oscillation frequency $f_{und} = \omega_{und}/(2\pi) = \sqrt{K}/(2\pi)$ vs. depth. (B) Far field resonant frequency $f_\sigma$ vs. depth. See section 6 in S3 and S4 Files for the calculations used for this plot.

gas makes no difference in either $\omega_{und}$ or $\omega_\sigma$ for any size of bubble in the two surface environments at different temperatures, and also makes no difference for large bubbles in deep water. For small bubbles (0.01 m radius) in deep water, $\omega_{und}$ is 6% higher and $\omega_\sigma$ is 5% higher with oxygen than with nitrogen.

## 4.3 Comparison between far-field resonant frequency and Minnaert frequency

Fig 3 shows the ratio of the far-field resonant frequency to the Minnaert frequency vs. depth. In shallow water the far-field resonant frequency is within 0.5% of the Minnaert frequency for all bubble radii and gases considered, but as the depth increases the far-field resonance frequency values diverge from the Minnaert frequency values. At depths of 1000 m and greater, the ratio of the far-field resonant frequency to the Minnaert frequency does not vary significantly with bubble radius (*cf.* section 7 in S3 and S4 Files). At depths of 1000 m, 2000 m, and 3000 m respectively the far-field resonance frequency values are 0.6%, 1%, and 2% greater than the Minnaert frequency for all bubble radii. Oxygen bubbles have far-field resonant frequencies that diverge from their Minnaert frequencies by slightly more than the far-field resonant frequencies of Nitrogen bubbles do.

## 4.4 Scaling with undamped frequency

Scaling relationships are useful when comparing systems with multiple parameters that vary by large amounts, as in comparisons of shallow-water bubbles to deep-water bubbles. Linear oscillators, such as a mass on a spring, scale according to their angular frequency of undamped

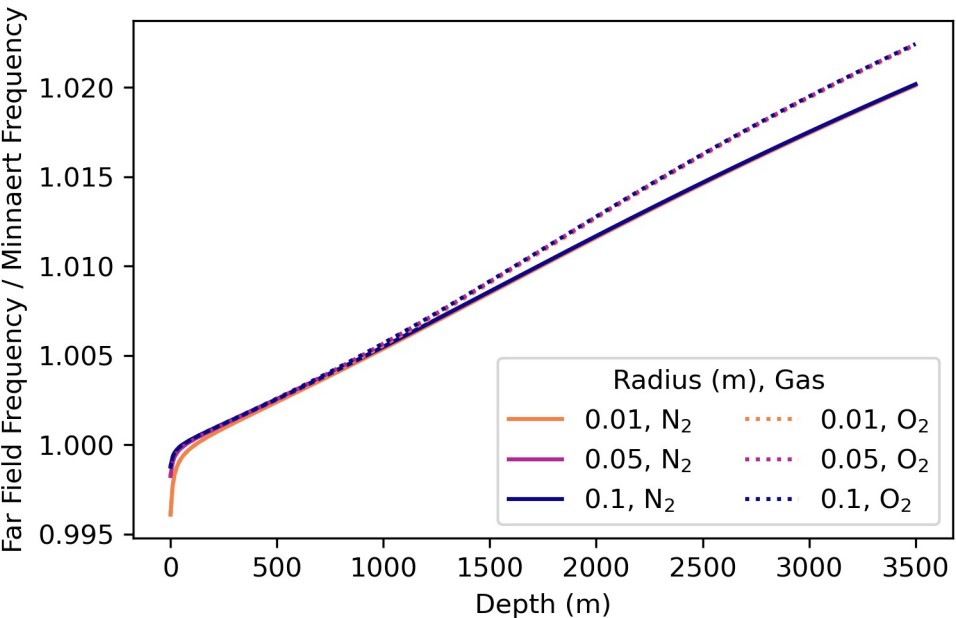

**Fig 3. Ratio of far-field resonant frequency to Minnaert frequency vs. depth.** The ratio of the far-field resonance frequency $f_\sigma$ to the Minnaert frequency $f_M$ vs. depth calculated for nitrogen and oxygen bubbles of radii 0.01 m, 0.05 m, and 0.10 m for a 1.5°C constant temperature, 35.0 g/kg constant salinity depth profile. Water and gas parameter values were calculated at 1 m depth intervals and interpolated between these depths. See section 7 in S3 and S4 Files for the calculations used for this plot.

oscillation, $\omega_{und} = \sqrt{k/m}$, where $k$ is the spring stiffness and $m$ the mass. The effect of the damping force on the motion of an undriven mass-spring system and both the transient and steady-state motion of the driven mass-spring system all depend on the relationships between $\omega$ and $\beta$ with $\omega_{und}$. Comparing Eqs (2) and (3), we use

$$\omega_{und} = \sqrt{K} \qquad (20)$$

as the scaling frequency for the driven bubble system. Oscillating bubbles are much more complicated than a mass-spring system because both $\omega_{und}$ and the damping factor $\beta$ have dependencies on the driving angular frequency $\omega$.

The angular frequency $\omega_{und}$ and the associated frequency $f_{und} = \omega_{und}/(2\pi)$ depend on the properties of the water, the bubble gas, and the bubble radius. Equation (20) has no closed-form solution for $\omega_{und}$, so we solve it numerically using the values of gas and water properties in each of the environments (Table 1) and bubble radii to obtain values of $\omega_{und}$ (*cf.* section 5.6 in S3 and S4 Files).

Over the range of parameters considered in this study, bubble oscillations and resonant frequencies scale with $\omega_{und}$. Table 2 shows values of the ratio $\omega_\sigma/\omega_{und}$ for various bubble radii to demonstrate the scaling of $\omega_\sigma$ with $\omega_{und}$. The ratio $\omega_\sigma/\omega_{und}$ varies by less than 0.006% from the mean value as the bubble radius increases from 0.01 m to 0.10 m in each of the environments (along the rows in Table 2). To simplify the frequency dependence of normalized parameters, the normalized frequency $f'$ is defined as

$$f' = \frac{f}{f_{und}} = \frac{\omega}{\omega_{und}}. \qquad (21)$$

**Table 2. Ratio of far-field resonance frequency to undamped frequency for selected bubble radii.**

| Bubble Gas | Environment | Bubble Radius (m) | | | | |
|---|---|---|---|---|---|---|
| | | 0.01 | 0.05 | 0.10 | 0.15 | 0.20 |
| $N_2$ | Warm Surface | 1.000042 | 0.999984 | 0.999974 | 0.999970 | 0.999967 |
| $O_2$ | Warm Surface | 1.000042 | 0.999984 | 0.999974 | 0.999970 | 0.999967 |
| $N_2$ | Cold Surface | 1.000033 | 0.999978 | 0.999969 | 0.999965 | 0.999963 |
| $O_2$ | Cold Surface | 1.000033 | 0.999978 | 0.999969 | 0.999965 | 0.999963 |
| $N_2$ | Depth 1000 m | 0.994444 | 0.994431 | 0.994428 | 0.994427 | 0.994426 |
| $O_2$ | Depth 1000 m | 0.994248 | 0.994235 | 0.994232 | 0.994231 | 0.994230 |
| $N_2$ | Depth 2000 m | 0.988162 | 0.988148 | 0.988144 | 0.988143 | 0.988142 |
| $O_2$ | Depth 2000 m | 0.987045 | 0.987029 | 0.987025 | 0.987023 | 0.987022 |
| $N_2$ | Depth 3500 m | 0.979400 | 0.979384 | 0.979381 | 0.979379 | 0.979378 |
| $O_2$ | Depth 3500 m | 0.977049 | 0.977031 | 0.977026 | 0.977025 | 0.977023 |

See section 8.4 in S3 and S4 Files for the calculations of these values.

We use the normalized frequency to compare bubble oscillations in environments in which resonances occur at different non-normalized frequencies.

## 4.5 Bubble resonance curves

To investigate bubble resonance of far-field pressure, we examine the scattering cross-section obtained from Eq (5) as it varies with driving angular frequency $\omega$. For comparison of different bubble radii, we normalize the scattering cross-sections by dividing them by the bubble surface area,

$$\sigma_s' = \frac{\sigma_s}{4\pi R_0^2},$$

(22)

where the denominator is the numerator of Eq (5). Fig 4 shows a graph of the normalized scattering cross-section $\sigma_s'$ vs. normalized frequency $f'$ for bubbles in the five environments with radii 0.01 m, 0.05 m, and 0.10 m.

Nitrogen and oxygen bubbles in both the warm and cold-surface environments have sharp far-field resonance peaks at normalized frequencies of $f' = 0.99998(2)$, which is very close to $f' = 1$, the frequency for undamped bubble oscillations. As the water depth increases the normalized frequencies of the far field resonance peaks decrease slightly. At a depth of 1000 m nitrogen bubbles have a resonance peak at $f' = 0.994430(4)$ and oxygen bubbles at $f' = 0.994234(5)$. At a depth of 2000 m nitrogen bubbles have a resonance peak at $f' = 0.988146(5)$ and oxygen bubbles at $f' = 0.987027(6)$. Finally, at a depth of 3500 m nitrogen bubbles have a resonance peak at $f' = 0.979382(6)$ and oxygen bubbles at $f' = 0.977029(6)$. The deep-water resonances for both gases are not as sharp as the shallow water peaks.

Bubbles in deep water are much less efficient at radiating sound power than comparable-size bubbles in shallow water. This reduction in radiation efficiency occurs because the resonance peaks of the bubble scattering cross-sections decrease with depth. See Fig 4. At a depth of 1000 m the peak scattering cross section is 80 times less than the peak in the cold shallow environment. At a depth of 2000 m the same peak is almost 200 times less, and at a depth of 3500 m the peak value has decreased by a factor of 300.

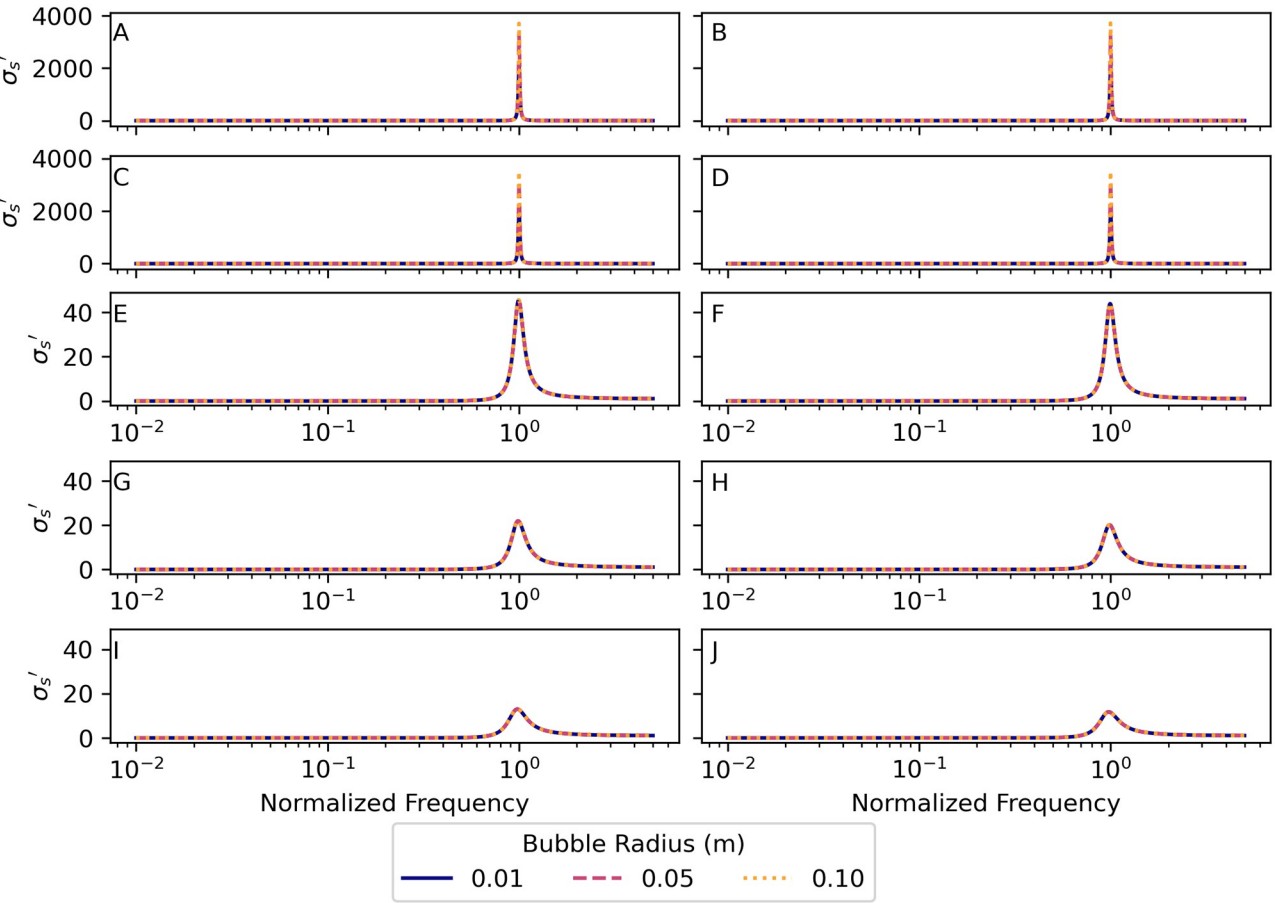

**Fig 4. Normalized scattering cross section $\sigma_s'$ vs. normalized frequency for bubbles.** Parameter values for each environment are given in Table 1. (A) Nitrogen bubbles in the warm-surface environment. (B) Oxygen bubbles in the warm-surface environment. (C) Nitrogen bubbles in the cold-surface environment. (D) Oxygen bubbles in the cold-surface environment. (E) Nitrogen bubbles in the 1000 m deep environment. (F) Oxygen bubbles in the 1000 m deep environment. (G) Nitrogen bubbles in the 2000 m deep environment. (H) Oxygen bubbles in the 2000 m deep environment. (I) Nitrogen bubbles in the 3500 m deep environment. (J) Oxygen bubbles in the 3500 m deep environment. Note that the vertical axis scales in subplots A–D are different than those in subplots E–J. The normalized scattering cross-section curves for all bubble radii in each environment are identical for each gas. See sections 8.1–8.3 in S3 and S4 Files for the calculations used for this plot.

## 4.6 Damping factor

Damping of bubble oscillations occurs due to three mechanisms that remove energy from the system: thermal losses $\beta_{th}$, viscous losses $\beta_{vis}$, and acoustic losses $\beta_{ac}$. Fig 5 shows graphs of the three contributions to the damping factor for bubbles of radius 0.10 m respectively in each of the environments. The viscous damping is negligible for all radii and environments considered. At low frequencies, thermal damping dominates, and at high frequencies acoustic damping dominates. The minimum value of the normalized damping factor occurs at the normalized frequency where the viscous and thermal damping factors are equal. This minimum-damping normalized frequency decreases with water depth.

The damping factor scales in both magnitude and frequency with the natural angular frequency $\omega_{und}$. To illustrate this, we define a normalized damping factor

$$\beta' = \frac{\beta}{\omega_{und}}. \tag{23}$$

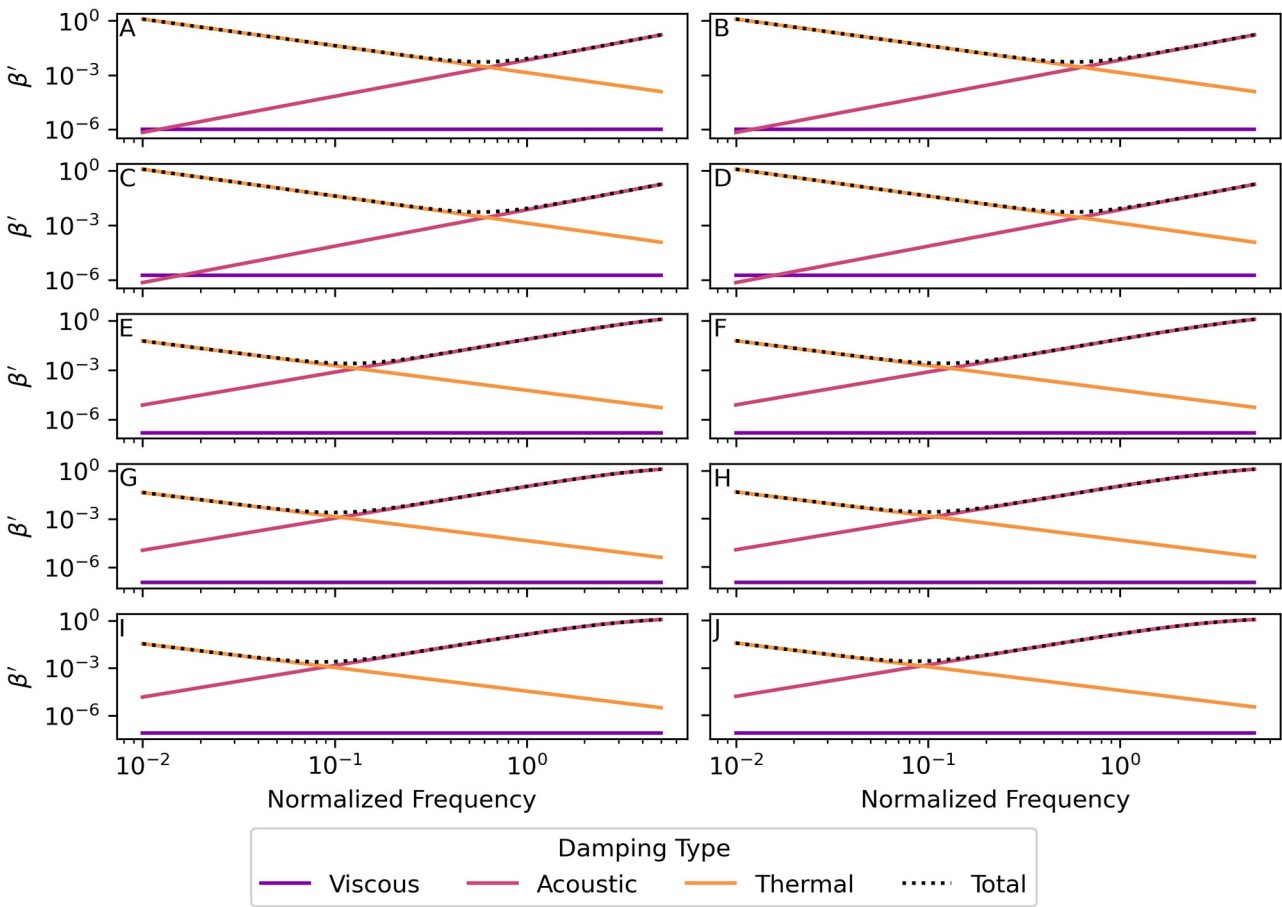

**Fig 5. Viscous, acoustic, and thermal contributions to the normalized damping factor $\beta'$.** The three parts of the normalized damping factor–viscous $\beta'_{vis} = \beta_{vis}/\omega_{und}$, acoustic $\beta'_{ac} = \beta_{ac}/\omega_{und}$, and thermal $\beta'_{th} = \beta_{th}/\omega_{und}$ damping–plotted vs. normalized frequency $f'$ for a bubble of radius 0.10 m. Parameter values for each environment are given in Table 1. (A) Nitrogen bubbles in the warm-surface environment. (B) Oxygen bubbles in the warm-surface environment. (C) Nitrogen bubbles in the cold-surface environment. (D) Oxygen bubbles in the cold-surface environment. (E) Nitrogen bubbles in the 1000 m deep environment. (F) Oxygen bubbles in the 1000 m deep environment. (G) Nitrogen bubbles in the 2000 m deep environment. (H) Oxygen bubbles in the 2000 m deep environment. (I) Nitrogen bubbles in the 3500 m deep environment. (J) Oxygen bubbles in the 3500 m deep environment. See sections 8.6 and 8.7 in S3 and S4 Files for the calculations used for this plot.

Fig 6 shows a graph of the normalized damping factor $\beta'$, vs. normalized frequency $f'$ for nitrogen and oxygen bubbles with radii 0.01 m, 0.05 m, and 0.1 m in the two surface and three deep-water environments. For bubbles of both gases in all environments, the minimum of the damping factor occurs at a frequency much lower than both the natural frequency of oscillation (normalized frequency $f' = 1$) and the resonant frequency (corresponding to the peaks in Fig 4). This indicates that variation in the damping factor is not the dominant factor in bubble resonance, although it may have a small effect on the value of the resonant frequency, which varies slightly near $f' = 1$.

At some driving frequencies the damping factor is sufficiently large that the system is overdamped. Oscillating systems [43] are underdamped when the damping factor is less than the natural angular frequency ($\beta/\omega_{und} < 1$), critically damped when the damping factor is equal to the natural angular frequency ($\beta/\omega_{und} = 1$), and overdamped when the damping factor is greater than the natural angular frequency ($\beta/\omega_{und} > 1$). Systems that are critically damped and overdamped do not oscillate but decay exponentially in amplitude unless driven by an oscillating force. The bubbles considered in this study are driven, so they could be forced into

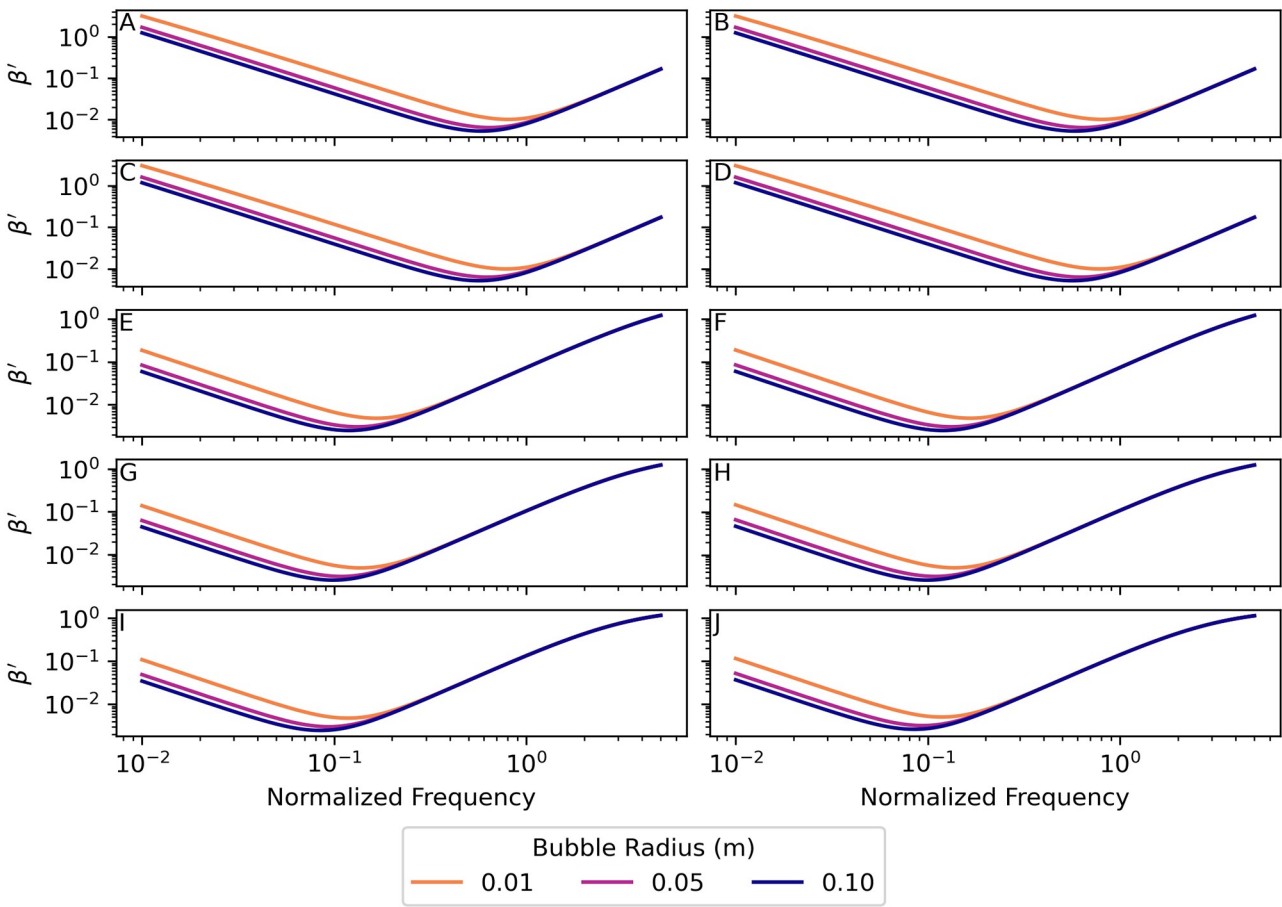

**Fig 6. Normalized damping factor $\beta' = \beta/\omega_{und}$ vs. normalized frequency $f'$ for several bubble radii.** Parameter values for each environment are given in Table 1. (A) Nitrogen bubbles in the warm-surface environment. (B) Oxygen bubbles in the warm-surface environment. (C) Nitrogen bubbles in the cold-surface environment. (D) Oxygen bubbles in the cold-surface environment. (E) Nitrogen bubbles in the 1000 m deep environment. (F) Oxygen bubbles in the 1000 m deep environment. (G) Nitrogen bubbles in the 2000 m deep environment. (H) Oxygen bubbles in the 2000 m deep environment. (I) Nitrogen bubbles in the 3500 m deep environment. (J) Oxygen bubbles in the 3500 m deep environment. See sections 8.8 and 8.9 in S3 and S4 Files for the calculations used for this plot.

oscillations, even when they are critically damped or overdamped. These oscillations would stop immediately and the amplitude would decay exponentially to zero when the driving force stops.

## 4.7 Quality factor

We calculate the quality factor $Q_f$ for bubbles of various radii in the shallow-water and in deep-water environments using Eq (18), with $\omega_\sigma$ as the resonant frequency and $\beta(\omega_\sigma)$ as the damping factor. Fig 7 is a graph of the quality factor vs. bubble radius for the five environments. Nitrogen bubbles in the warm shallow water environments have quality factors increasing from $Q_f = 46$ at $R_0 = 0.01$ m to $Q_f = 65$ at $R_0 = 0.20$ m. Oxygen bubbles in the warm shallow water environments have quality factors increasing from $Q_f = 46$ at $R_0 = 0.01$ m to $Q_f = 66$ at $R_0 = 0.20$ m. Nitrogen bubbles in the cold shallow water environments have quality factors that are slightly less than the warm shallow water values, increasing from $Q_f = 46$ at $R_0 = 0.01$ m to $Q_f = 63$ at $R_0 = 0.20$ m. Oxygen bubbles in the cold shallow water environments also have quality factors increasing from $Q_f = 46$ at $R_0 = 0.01$ m to $Q_f = 63$ at $R_0 = 0.20$ m. Bubbles of

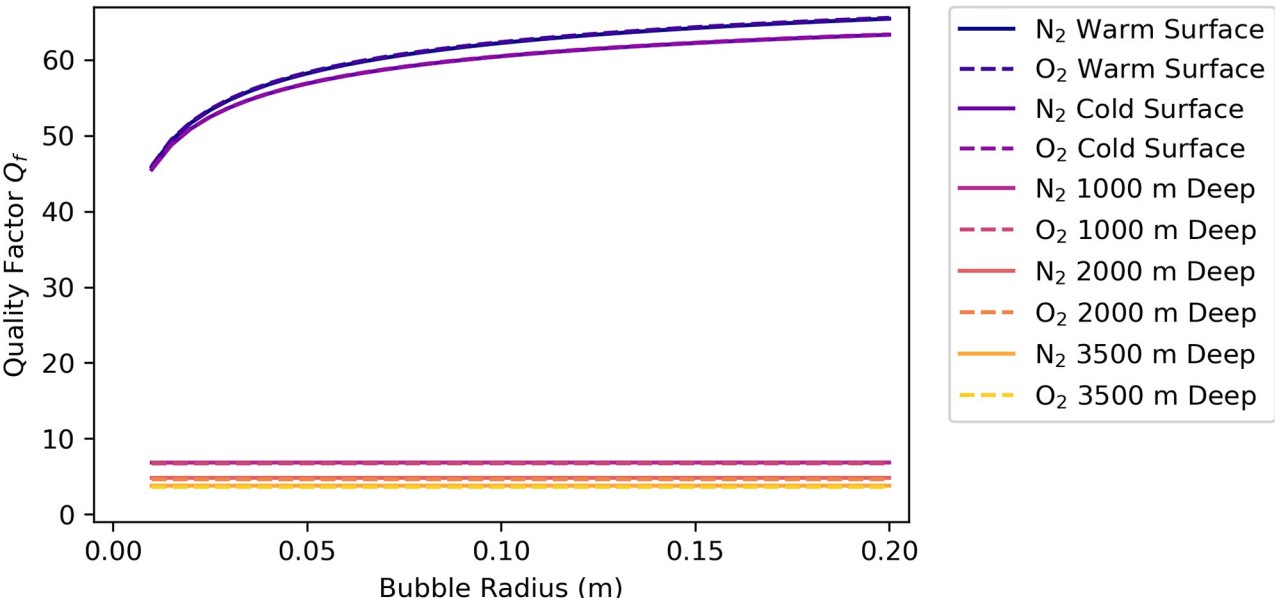

**Fig 7. Quality factor $Q_f$ vs. bubble radius.** Values are shown for bubbles at the warm ocean surface and cold ocean surface and at depths of 1000 m, 2000 m, and 3500 m. The quality factor curves for nitrogen and oxygen bubbles overlap in each of the environments. See section 9 in S3 and S4 Files for the calculations used for this plot.

both gases in each of the three deep-water environments have nearly constant quality factors as the bubble radius varies. Nitrogen bubbles have $Q_f = 6.8$ at 1000 m, $Q_f = 4.8$ at 2000 m, and $Q_f = 3.8$ at 3500 m. Similarly, oxygen bubbles have $Q_f = 6.7$ at 1000 m, $Q_f = 4.6$ at 2000 m, and $Q_f = 3.6$ at 3500 m. This decrease in quality factor with depth occurs because the damping factor at the resonant frequency $f' \approx 1$ increases with depth for all bubble sizes and gas types (see Fig 6) causing the resonances to decrease in peak value and increase in width (see Fig 4).

## 5 Discussion

### 5.1 Bubble resonance and fish sound production

Our analysis of driven bubbles in deep-water environments can be used to establish limits on the sound production mechanisms for deep-water fishes. The analysis of bubble oscillation also applies to fish hearing in auditory specialists [59] and target returns from sonar and echo sounders. In the 3500 m deep environment the temperature is low (1.5˚C), and the pressure is extremely high (35.62 MPa or 351.6 atm). The water density in this environment is about 1% higher than in the shallow-water environments. The sound speed in water a depth 3500 m falls between the warm-surface and cold-surface values because the decrease in sound speed due to the low temperature is offset by an increase in sound speed due to the high pressure. The other water properties are largely unaffected by the high pressure and have the same values as in the cold-surface environment, which has the same temperature as in deep water. The deep-water gas properties affecting bubble oscillations have very different values than in the surface environments. This means that the thermal and viscous losses for bubbles in the deep-water environment are higher than for same size bubbles in the shallow environments, while the acoustic losses–which depend on bubble size, oscillation frequency, and water sound speed [see Eq (12)]–are less affected by the extreme depth.

Nitrogen and oxygen gases have similar-value properties that affect bubble oscillations. As a result, there is very little difference between oscillations of bubbles of the two gases, and

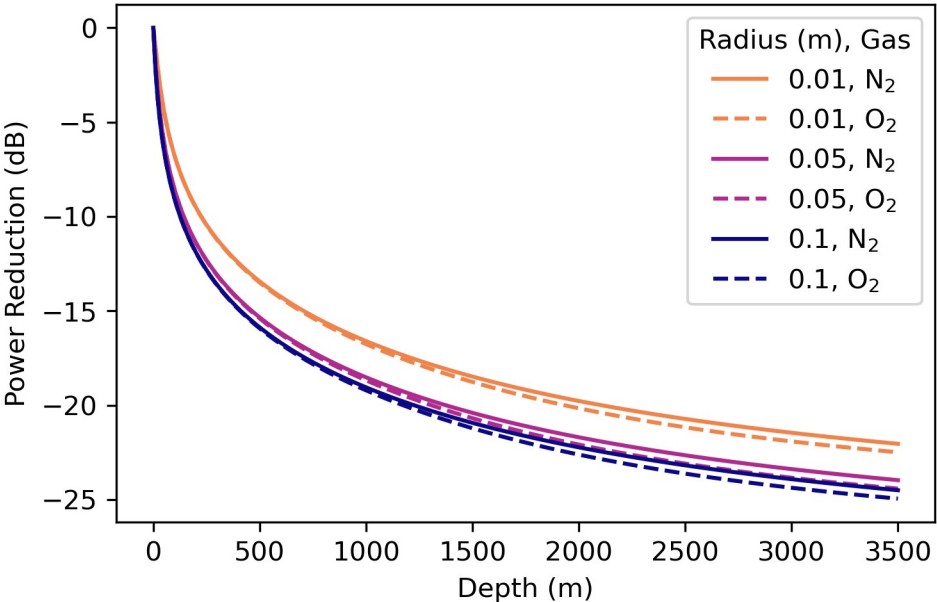

**Fig 8. Bubble radiated power reduction vs. depth.** This is calculated as the ratio of the scattering cross-section at depth to the surface scattering cross-section $\sigma_s(d)/\sigma_s(0)$ for a 1.5˚C constant temperature, 35.0 g/kg constant salinity depth profile. The decibel power reduction at depth $d$ is given by $10\log_{10}[\sigma_s(d)/\sigma_s(0)]$. See section 10 in S3 and S4 Files for the calculations used for this plot.

potential changes in gas content with depth are unlikely to affect sound radiation. The resonant frequencies for far-field pressure are much higher in the deep-water environments than for the same size bubbles in the surface environments, and the resonant frequency values for nitrogen and oxygen bubbles are almost the same [see Figs 1(B) and 2(B)].

The Minnaert frequency [1, 6] is a reasonable approximation for the far-field resonant frequency of swim-bladder-size bubbles in shallow water, but the far-field resonant frequency diverges from the Minnaert frequency with increasing depth. At a depth of 3500 m the far-field resonant frequency is approximately 2% greater than the Minnaert frequency (see Fig 3). This difference in predicted resonant frequencies is compounded by the increase of resonant frequencies with depth.

The scattering cross-sections (Fig 4) show that bubbles of both gases in deep water are less efficient radiators—by a factor of about 300 at depth 3500 m—than the same size bubbles in the shallow environments. This results in a reduction of about $10\log_{10}(300) = 25$ dB in source power level than the same-size bubble at the surface (the source sound pressure level is also reduced by 25 dB). Fig 8 is a graph of the ratio of the scattering cross-section at depth to the surface scattering cross-section $\sigma_s(d)/\sigma_s(0)$ for a 1.5˚C constant temperature, 35.0 g/kg constant salinity depth profile. Therefore, deep-water fishes might require powerful driving forces on the sound-producing organs (swim bladders) to overcome this inefficiency and still produce relatively quiet sounds. The scattering cross-section in the deep-water environment has a resonance peak that is less sharp and has a peak value that is much lower than the two surface environments for both gases, which is indicative of a higher-damped, lower quality factor $Q_f$ oscillator. The higher damping of the deep-water bubbles can be seen in Figs 5 and 6 by examining the values near normalized frequency $f' = 1$. A higher-damped, lower-value $Q_f$ system would be less sensitive to the precise frequency of the driving force, with a much more uniform response to frequencies above and below resonance.

Swim bladder wall tissue is viscoelastic [38] and adds both stiffness and damping to the bladder. The tissue in shallow-water fishes has high water content (e.g., 80% in toadfish [38]), and the water content of deep-water fish and perhaps their swim bladders could be even higher [60, 61]. As a result, the characteristic impedance of the wall tissue is essentially the same as the surrounding water and has negligible effect on mass loading. Near the surface, the added stiffness of the swim bladder wall tissue increases the resonant frequency of the swim bladder. However, as depth increases the stiffness of the gas becomes increasingly dominant so that the bladder wall contributes much less to the overall stiffness. The damping contribution of the swim bladder wall tissue decreases the resonance amplitude and the quality factor. The wall tissue is not affected by increasing pressure and depth, so its contribution to the overall damping is approximately constant. The net effect is that the swim bladder wall tissue likely has negligible effect on stiffness and resonant frequency at great depth but contributes additional damping and reduces the resonant amplitude and quality factor at all depths.

The shapes of fish swim bladders vary considerably, but most are better approximated by a prolate spheroid than by a sphere. Weston [5] shows that the natural frequency increases as a sphere deforms to a prolate spheroid, but that the effect is small. A major to minor semi-axis ratio of 2 gives about a 2% increase in frequency relative to a sphere of the same volume, and a ratio of 16 gives about a 30% increase.

Physical limitations on muscle contraction rates at low temperatures and the high natural frequencies of deep-water bubbles suggests that deep-water fishes are likely to produce pulsed sounds rather than steady-state sounds. One possibility for overcoming the need for high muscle contraction rates is a mechanism that loads the swim bladder slowly and releases suddenly producing a burst of transient sound. Such a system is likely to have dipole or even quadrupole contributions depending on the configuration of the excitation resulting in a directivity pattern that is not isotropic [29, 62].

Our analysis uses theoretical and numerical calculations to predict bubble and swim bladder properties because empirical measurements in the extremes of the deep-water environments are difficult to obtain. Deep-water conditions could be replicated in a high-pressure chamber. Another possibility for obtaining empirical measurements is to use a deep-diving submarine or drone to make measurements at depth in the ocean. To our knowledge, such measurements of bubble properties in extreme deep-water conditions have never been attempted.

## 5.2 Biological considerations

Sound is often considered an excellent medium for communication underwater due to rapid velocity and long-distance propagation. However, the almost complete absence of known fish sounds from the continental slope and abyss is a glaring omission, considering the deep ocean is the largest habitat on earth [63, 64]. In fact, in a recent glider study, Wall *et al.* [65] noted quiet in deeper waters suggesting reduced fish sounds on the continental shelf break and upper slope waters off the Southeastern United States. In addition to the low density of fishes on the deep slope [64, 66], obvious impediments to recording such sounds would include difficulties supplying suitable instrumentation at the right place and time. Further, providing lights to identify any potential callers in long-term installations [23] might inhibit courtship behavior.

Recent work on sonic swim bladder systems in neobythitine cusk-eels (Ophidiidae) from the continental slope [18, 19] stimulated us to focus on the effect of hydrostatic pressure and temperature on sound emission using mathematics to predict the behavior of an underwater resonant bubble, a classic if imperfect model of swim bladder acoustics [2, 3]. Our finding of a

rapid drop-off in sound power with depth likely contributes to the myriad difficulties in obtaining successful recordings. In addition to the higher damping and 25 dB decrease in power for an underwater bubble at 3500 m, the viscoelastic swim bladder wall [38], responsible for rapid damping of fish sounds [27] will further decrease sound radiation.

The single published recording likely from an unknown fish from the upper slope [20] may at first appear to contradict our thesis of quieter sounds in deep water. Possibly from an ophidiid, it was estimated by triangulation to come from a depth of 548 m to 696 m at a distance of ~4000 m from the hydrophone array with a source level of 131(3) dB re 1 μPa Hz$^{-1/2}$ measured on a frequency band from 800 Hz to 1000 Hz and assuming spherical spreading. Integrating this source level over the spectral band gives an effective source RMS sound pressure level of 154(3) dB re 1 μPa.

Source levels of several continental-shelf and estuarine fishes have been measured *in situ*, including some that are much louder than the deep-water fish detected by Mann and Jarvis [20]. Parsons *et al.* [67] used spherical spreading to estimate a source sound pressure level of 170 dB re 1 μPa for mulloway *Argyrosomus japonicus*, a large sciaenid. Locascio and Mann [68] measured source levels of black drum *Pogonias cromis*, also a large sciaenid, at 165(1) dB re 1 μPa. Sprague and Luczkovich [69] used spherical spreading to estimate the source sound pressure level of a silver perch *Bairdiella chrysoura*, a small sciaenid, at 128 dB to 135 dB re 1 μPa. The oyster toadfish *Opsanus tau*, a small batrachoidid, has a source level of 130 dB re 1 μPa [29], and dense choruses of the small, shallow-water, bottom-dwelling ophidiid striped cusk-eel *Ophidium marginatum* have been measured with a 60 s energy flux density of 150 dB re 1 μPa$^2$s in water depths of 12 m [70]. This is equivalent to an average received sound pressure level of 132 dB re 1 μPa. The unknown deep-water fish detected by Mann and Jarvis [20] produced sounds with source levels that are 11 dB to 16 dB quieter than the larger continental shelf species, which is consistent with the source level reduction predicted by the curve in Fig 8 for depths close to 600 m and with our calculation of a 25 dB reduction in source level at a depth of 3500 m.

To put the 154 dB re 1 μPa source level from the upper slope into perspective, we compare relative development of the sonic system of three upper-slope species *Hoplobrotula armatus*, *Neobythites longipes* and *N. unimaculatus* collected at 200 m to 300 m [18] with *Dicrolene intronigra* collected from depths close to 1 km and deeper [19]. *Dicrolene* has been captured from 670 m to 1700 m [71]. All four of these species have muscle pairs that connect directly or indirectly to the swim bladder. The ventral sonic muscles that connect directly to the swim bladder are considerably larger in males than in females. Notably the swim bladder and sonic muscles of male *Dicrolene* are heavier (approximately by a factor of 2) than in the three shallower species [19] suggesting sufficient food at a depth of 1 km or more to maintain a robust sonic system. In fact the larger muscles and swim bladder could potentially compensate for weaker sound generation at a depth of 1 km.

Species at greater depths have to contend not only with the increasing loss in sound power under high pressure but with a nutritionally dilute environment [72–75]. These fishes tend have long tails, move with an eel-like anguilliform motion, have high water content, reduced metabolism [76–78] and likely little spare energy to devote to a robust sonic system. *Acanthonus armatus*, a member of the neobythitine subfamily, has a minute brain, consists of almost 90% water and has lost its swim bladder [60, 61]. Two deep neobythitines (*Porogadus miles* captured between 1900 m, and 2170 m and *Bathyonus pectoralis* captured between 3422 m, and 5000 m) have smaller bladders with thinner more pliable walls than upper-slope species and have reduced their sonic muscles to a single muscle pair connecting to the medial bladder [19]. Muscle weight was still considerably greater in males than in females likely indicating their importance in male courtship. Notably these muscles terminate considerably before the

bladder, ending in a long tendon, particularly in the deeper species *Bathyonus* in which the tendon took up 70% of the distance between the muscle origin and the swim bladder. The tendon was interpreted as an adaptation to allow rapid bladder movement for sound production while using slower muscle contractions and less energy [79, 80]. The tendon elasticity would presumably be impacted minimally by increased pressure and allow greater bladder deformation with less energy expenditure although equal bladder movement would require more force under higher hydrostatic pressures. Larger muscle fibers within the larger muscle in male *Bathyonus* likely provide a greater force [19]. Another deep species *Barathodemus manatinus* [81] has the medial muscle present only in males but without a long tendon. Unfortunately, the sonic systems of few deep water cusk-eels have been examined.

From dives in deep-water submersibles, Ken Sulak (personal communication) notes that he has rarely seen more than one fish at a time at depths below 1500 m, which makes finding mates particularly challenging. We see several adaptations for sound production in deep-water neobythitines ranging from *Acanthonus armatus*, which has lost its swim bladder [60] to *Barathodemus manatinus* with a single pair of sonic muscles that attach to the swim bladder in males but not females [81] to *Porogadus miles* and *Bathyonus pectoralis* that have short medial muscle pairs that attach to the swim bladder via long tendons. Clearly *Acanthonus* is mute, and we speculate that *Barathodemus* would produce weak sounds in close proximity to a female it is already courting. Perhaps with their long tendons, *Porogadus* and *Bathyonus* can overcome pressure effects and broadcast sounds of sufficient amplitude to attract females from somewhat greater distances. The findings of this study coupled with the low density of fishes on the deep continental slope help explain the absence of recorded fish sounds.

## 6 Summary and conclusions

Our analysis predicts the far-field resonant frequency, damping factor, and quality factor that should be useful to a number of fields in fish bioacoustics. Using a spherical bubble as the primary model, we solve the mechanical and thermodynamic equations for the far field resonant frequency and the damping for five ocean environments–warm and shallow, cold and shallow, and cold environments at depths of 1000 m, 2000 m and 3500 m–where the cold-shallow and cold-deep environments are the same temperature in order to isolate thermal effects. We also look at the effects of bubble size, and nitrogen and oxygen as contained gases. We make several observations from our calculations. The undamped natural frequency and far field resonant frequency increase with decreasing bubble radius in all environments. The temperature difference in the shallow water environments has virtually no effect on either the far field resonant frequency or the damping. Differences in bubble gas also have little effect on either far field resonant frequency or damping in either shallow or deep water. For small bubbles, oxygen produces a slightly higher increase in resonant frequency with depth, but at 3500 m the difference in the far field resonant frequency between nitrogen and oxygen is only 5%; therefore, potential changes in gas content with depth will have minimal effect on fish hearing and sound production. The undamped natural frequency and the far field resonant frequency (which maximizes the scattering cross-section) are virtually identical for any given condition (*i.e.*, bubble size, temperature and depth have the same effect on both). The damping factor at the resonant frequency increases with depth resulting in lower and wider resonance peaks in the scattering cross-section and a lower quality factor. The dominant effects of increasing depth are:

1. The far field resonant frequency increases due to increasing hydrostatic pressure and gas stiffness.

2. The damping factor at the resonant frequency increases. The increase is primarily due to thermal and viscous losses. Acoustic losses are less affected by increasing depth.

3. The scattering cross-section (ratio of far field radiated power to input power) decreases by a factor of 300 (corresponding to a 25 dB loss in radiated sound power at depth 3500 m) due to the increased damping losses at the resonant frequency.

4. The quality factor decreases with depth, resulting in broad-frequency scattering cross-section curves in deep water suggesting that deep-water swim bladders are less sensitive to driving frequency than shallow-water swim bladders.

In addition to the quantitative analysis of spherical bubbles, we argue qualitatively that the viscoelasticity of the swim bladder wall contributes additional viscous losses that further increase the damping and decrease the radiated power (scattering cross-section). The non-spherical shapes of swim bladders also tend to increase their natural frequencies, although this effect is minor unless there is extreme elongation of the swim bladder. The high far-field resonant frequencies of smaller bubble sizes in the deep water environment are above the maximum frequencies that can be detected by most fishes, suggesting that deep-water fishes either do not use swim bladder resonance to produce sound or have swim bladders with radii larger than ∼0.05 m. Deep water fishes most likely produce sound by forced muscle action, either with intermittent pulses or short continuous oscillations rather than by swim bladder resonance. Due to swim bladder shapes and the geometry of muscles, tendons and ligaments, sounds are also likely to have a more directional radiation pattern [29, 62] than the omnidirectional pattern produced by an oscillating spherical bubble.

We propose that the reasons there are no recorded fish sounds from ocean depths greater than about 700 m are the low population density of fishes in deep habitats, the increase in the far-field resonant frequency and damping of swim bladder oscillations with increasing depth, and the concomitant reduction in the resonance quality factor and scattering cross-section leading to sound production at lower acoustic power than a similar size swim bladder near the surface.

## Supporting information

**S1 File. Jupyter notebook with calculations of seawater and gas properties.** Calculations use the Julia Programming Language [44] version 1.7.2.
(IPYNB)

**S2 File. PDF containing seawater and gas property calculations.** PDF file created from S1 File. This file does not require Jupyter or Julia for viewing.
(PDF)

**S3 File. Jupyter notebook with calculations of bubble dynamics.** Calculations use the Julia Programming Language [44] version 1.7.2.
(IPYNB)

**S4 File. PDF containing bubble dynamics calculations.** PDF file created from S3 File. This file does not require Jupyter or Julia for viewing.
(PDF)

## Author Contributions

**Conceptualization:** Mark W. Sprague, Michael L. Fine, Timothy M. Cameron.

**Data curation:** Mark W. Sprague.

**Formal analysis:** Mark W. Sprague.

**Investigation:** Mark W. Sprague.

**Methodology:** Mark W. Sprague.

**Software:** Mark W. Sprague.

**Validation:** Michael L. Fine, Timothy M. Cameron.

**Visualization:** Mark W. Sprague.

**Writing – original draft:** Mark W. Sprague, Michael L. Fine, Timothy M. Cameron.

**Writing – review & editing:** Mark W. Sprague, Michael L. Fine, Timothy M. Cameron.

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
