## [Decision Letter · Decision Letter 0]

23 May 2022

PONE-D-22-10057An investigation of bubble resonance and its implications for sound production by deep-water fishesPLOS ONE

Dear Dr. Sprague,

Thank you for submitting your manuscript to PLOS ONE. After careful consideration, we feel that it has merit but does not fully meet PLOS ONE’s publication criteria as it currently stands. Therefore, we invite you to submit a revised version of the manuscript that addresses the points raised during the review process.

We look forward to receiving your revised manuscript.

Kind regards,

Mohammad Mehdi Rashidi

Academic Editor

PLOS ONE

Journal Requirements:

Additional Editor Comments:

Dear Authors,

I strongly advise you just cite the related papers in the literature review. In this case, just one or two papers (completely related) would be enough. Please ignore the suggested papers that cannot improve literature review significantly.

Editor

Reviewers' comments:

Reviewer's Responses to Questions

**Comments to the Author**

1. Is the manuscript technically sound, and do the data support the conclusions?

Reviewer #1: Yes

Reviewer #2: Yes

Reviewer #3: Yes

2. Has the statistical analysis been performed appropriately and rigorously? 

Reviewer #1: Yes

Reviewer #2: N/A

Reviewer #3: Yes

3. Have the authors made all data underlying the findings in their manuscript fully available?

Reviewer #1: Yes

Reviewer #2: Yes

Reviewer #3: Yes

4. Is the manuscript presented in an intelligible fashion and written in standard English?

Reviewer #1: Yes

Reviewer #2: Yes

Reviewer #3: Yes

5. Review Comments to the Author

Reviewer #1: Hi dear editor

The novelty and presentation of the paper is enough, and also the manuscript has a good structure. I recommended the msnuscript for publication without any comments.

Best regards

Sh. Shoeibi

Reviewer #2: The paper concerns two parts; 1) a highly technical part on bubble physics and environmental impacts on bubble acoustic properties and 2) a well-written part on biological interpretation and addressing relevant data from the literature. I am happy to believe the first part and was pleased to read about the second part. The authors wrote an excellent discussion about the relevance and insights about sound production in fishes in deep water layers. It would be nice if they would address the conditions for various depths (2000 is also already very deep) also in the results in a graph or table, besides the extreme conditions of 3500 m. I also believe the authors could expand a bit on how they believe this field of study could make additional progress and how to gather empirical data.

Reviewer #3: Dears,

In this manuscript, the authors adopted an investigation of bubble resonance and its implications for sound production by deep-water fishes. The paper is interesting, however, there are a few changes that should be made as follows:

a. Theoretical section:

a- 1. In the abstract, please explain and discuss the “novel” word.

a- 2. Details of main and boundary condition flows should be mentioned in the table such as flow rate, temperature, and,… .

a- 3. An algorithm and schematic form of numerical solution should be added.

a- 4. The literature review need to improve, which may also require a more substantial literature review (Especially in modeling of droplets such as https://doi.org/10.1016/j.ijheatmasstransfer.2021.122392, ....).

b. Format of the manuscript:

b- 1. The manuscript should be polished in both language and structure.

6. PLOS authors have the option to publish the peer review history of their article (what does this mean?). If published, this will include your full peer review and any attached files.

Reviewer #1: No

Reviewer #2: No

Reviewer #3: No

---

## [Author Response · Author response to Decision Letter 0]

14 Jun 2022

All reviewer comments are addressed in the Response to Reviewers document.

---

## [Decision Letter · Decision Letter 1]

23 Jun 2022

An investigation of bubble resonance and its implications for sound production by deep-water fishes

PONE-D-22-10057R1

Dear Dr. Sprague,

We’re pleased to inform you that your manuscript has been judged scientifically suitable for publication and will be formally accepted for publication once it meets all outstanding technical requirements.

Kind regards,

Mohammad Mehdi Rashidi

Academic Editor

PLOS ONE

Additional Editor Comments (optional):

Reviewers' comments:

Reviewer's Responses to Questions

**Comments to the Author**

1. If the authors have adequately addressed your comments raised in a previous round of review and you feel that this manuscript is now acceptable for publication, you may indicate that here to bypass the “Comments to the Author” section, enter your conflict of interest statement in the “Confidential to Editor” section, and submit your "Accept" recommendation.

Reviewer #1: All comments have been addressed

Reviewer #3: All comments have been addressed

2. Is the manuscript technically sound, and do the data support the conclusions?

Reviewer #1: Yes

Reviewer #3: Yes

3. Has the statistical analysis been performed appropriately and rigorously? 

Reviewer #1: Yes

Reviewer #3: Yes

4. Have the authors made all data underlying the findings in their manuscript fully available?

Reviewer #1: Yes

Reviewer #3: Yes

5. Is the manuscript presented in an intelligible fashion and written in standard English?

Reviewer #1: Yes

Reviewer #3: Yes

6. Review Comments to the Author

Reviewer #1: The authors did well revised in the manuscript and the manuscript accepted and is ready for publication.

Reviewer #3: (No Response)

7. PLOS authors have the option to publish the peer review history of their article (what does this mean?). If published, this will include your full peer review and any attached files.

Reviewer #1: No

Reviewer #3: No

---

## [Editor Report · Acceptance letter]

29 Jun 2022

PONE-D-22-10057R1 

An investigation of bubble resonance and its implications for sound production by deep-water fishes 

Dear Dr. Sprague:

I'm pleased to inform you that your manuscript has been deemed suitable for publication in PLOS ONE. Congratulations! Your manuscript is now with our production department. 

Kind regards, 

on behalf of

Professor Mohammad Mehdi Rashidi 

Academic Editor

PLOS ONE